# Kaposi’s Sarcoma: Evaluation of Clinical Features, Treatment Outcomes, and Prognosis in a Single-Center Retrospective Case Series

**DOI:** 10.3390/cancers16040691

**Published:** 2024-02-06

**Authors:** Irene Russo, Dario Marino, Claudia Cozzolino, Paolo Del Fiore, Fitnete Nerjaku, Silvia Finotto, Annamaria Cattelan, Maria Luisa Calabrò, Anna Belloni Fortina, Francesco Russano, Marcodomenico Mazza, Sara Galuppo, Elisabetta Bezzon, Marta Sbaraglia, Marco Krengli, Antonella Brunello, Simone Mocellin, Stefano Piaserico, Mauro Alaibac

**Affiliations:** 1Soft-Tissue, Peritoneum and Melanoma Surgical Oncology Unit, Veneto Institute of Oncology IOV – IRCCS, 35128 Padova, Italy; irene.russo@iov.veneto.it (I.R.); claudia.cozzolino@studenti.unipd.it (C.C.); francesco.russano@iov.veneto.it (F.R.); marcodomenico.mazza@iov.veneto.it (M.M.); simone.mocellin@unipd.it (S.M.); 2Oncology 1 Unit, Department of Oncology, Veneto Institute of Oncology IOV – IRCCS, 35128 Padova, Italysilvia.finotto@iov.veneto.it (S.F.);; 3Department of Cardiac, Thoracic, Vascular Sciences, and Public Health, University of Padova, 35128 Padova, Italy; 4Department of Medicine (DIMED), School of Medicine, University of Padova, 35128 Padova, Italy; fitnete.nerjaku@gmail.com (F.N.); marta.sbaraglia@unipd.it (M.S.); 5Infectious and Tropical Diseases Unit, Padova University Hospital, 35128 Padova, Italy; annamaria.cattelan@unipd.it; 6Immunology and Molecular Oncology, Veneto Institute of Oncology IOV – IRCCS, 35128 Padova, Italy; luisella.calabro@iov.veneto.it; 7Dermatology Unit, Department of Medicine, University of Padova, 35128 Padova, Italy; anna.bellonifortina@unipd.it (A.B.F.); stefano.piaserico@unipd.it (S.P.); mauro.alaibac@unipd.it (M.A.); 8Pediatric Dermatology Regional Center, Department of Women’s and Children’s Health, University of Padova, 35128 Padova, Italy; 9Radiotherapy Unit, Veneto Institute of Oncology IOV – IRCCS, 35128 Padova, Italy; sara.galuppo@iov.veneto.it (S.G.); marco.krengli@unipd.it (M.K.); 10Radiology Unit, Veneto Institute of Oncology, IOV – IRCCS, 35128 Padova, Italy; 11Department of Pathology, Azienda Ospedale—University of Padova, 35128 Padova, Italy; 12Department of Surgery, Oncology and Gastroenterology (DISCOG), University of Padova, 35128 Padova, Italy

**Keywords:** Kaposi’s sarcoma, vascular sarcoma, iatrogenic disease, HIV-related disease, immunosuppression status

## Abstract

**Simple Summary:**

Kaposi’s sarcoma (KS) is a low-grade, vascular tumor associated with human herpesvirus 8 (HHV8) infection. There are four widely recognized types of KS including classic, endemic, iatrogenic, and epidemic forms. Although in most cases, KS is an indolent disease limited to the skin, it may have a more aggressive behavior resulting in locally aggressive disease and/or involving the mucosae or visceral organs, especially in immunosuppressed patients. The main risk factors are HHV-8 infection and immunosuppression status. The treatment is very heterogenous, because there is still no unified strategy for the treatment of KS. The aim of this study is to describe the demographic and clinical features, treatment outcomes, and prognosis of a cohort of patients affected by KS treated at the University Hospital of Padua (AOPD) and at the Veneto Institute of Oncology (IOV) between 1993 and 2022.

**Abstract:**

Kaposi’s sarcoma (KS) is a rare angioproliferative tumor classified in four different clinical–epidemiological forms. The diagnosis is based on histopathological and immunohistochemical analyses. The treatment is heterogeneous and includes several local and systemic therapeutic strategies. Methods: This is a retrospective cohort study including 86 KS patients treated between 1993 and 2022 at the University Hospital of Padua (AOPD) and at the Veneto Institute of Oncology (IOV). The data were extracted from an electronic database. Survival curves were generated using the Kaplan–Meier method, and Cox regression models were employed to explore associations with overall and disease-free survival. The male sex (89.53%), classical variant (43.02%), and cutaneous involvement (77.9%) were predominant. More than 61.6% of patients received a single treatment. Surgery, antiretroviral therapy, and chemotherapy were the mostly adopted approaches. A persistent response was observed in approximately 65% of patients, with a 22% relapse rate (at least 2 years). The overall survival ranges from 90 to 70% at 2 to 10 years after the diagnosis. Iatrogenic KS demonstrated a higher mortality (52.9%). This study reflects our experience in the management of KS. Comorbidities are very frequent, and treatments are heterogeneous. A multidisciplinary approach involving multiple referral specialists is essential for the appropriate management of this disease during diagnosis, treatment, and follow-up.

## 1. Introduction

Kaposi’s sarcoma (KS) is a rare angioproliferative tumor usually arising in the skin. Although in most cases, KS is an indolent disease limited to the skin, it may have a more aggressive behavior resulting in locally aggressive disease and/or involving the mucosae or visceral organs, especially in immunosuppressed patients. 

Over the years, four different clinicoepidemiological forms of KS have been identified: a sporadic form with an indolent course, affecting elderly men from the Mediterranean area (classical KS) [1,2]; a more aggressive form of KS, which affects young adults and children in sub-Saharan Africa (endemic KS) [3]; a form observed in transplant recipients and patients receiving aggressive immunosuppressive therapies (iatrogenic KS) [4]; and finally, a form reported in HIV-positive patients (epidemic KS) [5,6].

The data from the RARECARENet project report an incidence rate in Europe of 0.28 per 100,000 people/year [7]. The incidence in men is four times higher than in women. The main risk factors are human herpesvirus 8 (HHV-8) infection and immunosuppression status [8,9]. 

The diagnosis of KS is based on histopathological and immunohistochemical analyses. The treatment is heterogeneous and includes local and/or systemic therapeutic strategies [9].

Our study aimed to describe the characteristics of patients with KS, who were diagnosed and treated at the University Hospital of Padova (AOPD) and at the Veneto Institute of Oncology (IOV) between 1993 and 2022. The goal was to describe the demographic and clinicopathological characteristics of patients, disease features, and the diagnostic–therapeutic pathway through the creation of a unique database with the aim of better characterizing the disease, expanding knowledge, and providing better patient care.

## 2. Materials and Methods

### 2.1. Study Design

This is a retrospective cohort study of patients diagnosed and/or treated for KS between 1993 and 2022 at University Hospital of Padova (AOPD) and at the Veneto Institute of Oncology (IOV), which represent level III referral centers in the northeast of Italy.

### 2.2. Patients

The study included 86 patients. Most patients came for diagnosis and/or first-line treatment, while some patients came to our centers for disease progression assessments after being treated in local level II centers (Figure 1 and Figure 2).

### 2.3. Diagnosis and Treatment

KS was diagnosed using histopathological and immunohistochemical analyses from the biopsy of the lesion. Diagnostic and therapeutic clinical decisions resulted from a multidisciplinary collaboration between different specialists including dermatologists, infectious disease specialists, oncologists, radiation oncologists, radiologists, pathologists, and surgeons. A follow-up was carried out using contrast-enhanced CT scanning and lymph node and soft tissue ultrasounds. The RECIST (response evaluation criteria in solid tumors) criteria were used to assess the disease progression or response to treatment [10]. 

### 2.4. Data Collection

All data were extracted from an electronic database purposely built for this study. Patients were divided into four groups according to the clinical–epidemiological type of disease: classic, endemic, epidemic, and iatrogenic. The demographic information included the age at diagnosis, sex, and race, while information regarding the tumor included the anatomic site, involvement, and histopathologic features. Patients’ comorbidities were assessed, and neoplastic and autoimmune comorbidities were evaluated separately. Therapies included medical, surgical, and radiotherapy treatments and were classified as single or multiple treatments. The follow-up data were extracted during scheduled visits.

### 2.5. Statistical Analysis

Continuous data were summarized as the median and interquartile range (IQR). Disease-free survival (DFS) was calculated from the date of diagnosis to the date of the event or last follow-up visit, where the event was a relapse after initial treatment or other recurrences of disease. Survival curves were estimated with the Kaplan–Meier method, and (overall survival and disease-free survival) hazard ratios (HR) were calculated with the Cox regression model. All models are intended to be univariate, with the exception of the tumor site and treatments, for which a multivariable regression was fitted. Reciprocal HR, 1/HR, and <1 indicate a worse survival rate than the reference class, >1 indicates better. Values of *p* < 0.10 were formatted in bold, and results with *p*-value < 0.05 were considered statistically significant. The statistical analyses were performed using R 4.1 (R Foundation for Statistical Computing, Vienna, Austria) [11].

### 2.6. Ethics Considerations

The study was conducted in accordance with the principles of the 1975 Declaration of Helsinki, and patients gave their informed consent to the collection of their data for scientific purposes. The study was approved by the local ethics committee. 

## 3. Results

The study included 86 patients, whose demographic, tumor, clinical, and treatment characteristics are reported in Table 1.

The age of our patients ranged from 13.9 to 85.6 years, with a median age at diagnosis of 57.2 years and a mean age at diagnosis of 56.9 years. Of the 86 cases, 77 were men and only 9 women, with a male/female ratio of 8.5:1. The majority of patients (61.6%) had a single anatomical site involved, while 38.3% had multiple sites involved. The most frequently affected anatomical site was the lower limbs, followed by the head and neck, upper limbs, and trunk.

Based on the clinical–epidemiological type, the subjects were distributed into four groups. The classical variant was the most frequent (43%), followed by epidemic (33.7%), iatrogenic (19.7%), and endemic (3.4%) KS. 

The majority of patients presented only with cutaneous involvement (77.9%) while 5.8% had an exclusively extracutaneous presentation of the disease. Patients presenting with both cutaneous and extracutaneous involvement were 12.2%. 

Epidemic KS, thus related to HIV-1 and 2, constituted a considerable percentage in our study (34.8%). 

Of the iatrogenic type, 76.4% were organ transplant patients (with the liver being the most frequently transplanted organ).

In terms of comorbidities, 86% of the patients were affected by other diseases, including autoimmune (19.7%) and neoplastic (25.5%) disorders. The HHV-8 status was established in 55.8% of cases and was positive in 94.1% (48/51) of patients tested.

### 3.1. Treatment

About 61.6% of the patients received only one treatment, while in 38.3% of the cases, it was necessary to use several treatments combined in various ways. Treatment approaches were both local and systemic and included antiretrovirals, immunosuppression dose reduction, topical application of imiquimod, interferon therapy, cryotherapy, chemotherapy, surgery, electrochemotherapy, and radiotherapy (Figure 3). Twenty patients were treated with chemotherapy: among these, 5 patients also underwent second- or third-line therapies. Thirteen patients received liposomal doxorubicin as a first-line treatment, and six patients received gemcitabine (three as first-line and three as second-line therapies). Paclitaxel was administered to four patients (first-line therapy only in one case). The two other patients received a combination of vinorelbine and bleomycin or doxorubicin, bleomycin and vinorelbine respectively. Another patient received etoposide as a third-line therapy. 

Five patients were treated with electrochemotherapy alone and were all categorized as KS classic type.

### 3.2. Response and Relapse

A persistent response (for at least 2 years) was achieved in about 65% of patients. Seventeen patients died from the following causes: respiratory, liver, or kidney failure, infection, neoplasm, cardiac complications, and old age. 

The median follow-up was of 5.8 years, while the median interval to disease recurrence was 2.7 years. The median time interval between the diagnosis of KS and death occurrence was 2.4 years.

Relapse occurred in about 22% of cases. A total of 35.1% of patients with classic KS, 33.3% of those with the endemic type and 17.2%. of those with the epidemic type had a relapse. No patients with iatrogenic KS had a relapse (Figure 4A).

The mortality was higher in patients affected by iatrogenic KS (52.9%), followed by the endemic type (33.3%), classic (10.8%), and epidemic type (10.3%) (Figure 4B).

### 3.3. Survival Analysis

The overall survival (OS) at 2.5 years after diagnosis of KS was about 90%, while it was about 80% at 5 years. The tendency of the curve is to decrease very gradually after the first 2.5 years (Figure 5A).

The OS at 2.5 years was almost 100% in patients with a persistent response (of at least two years) and 70% in the group without a persistent response (Figure 5B). The disease-free survival (DFS) at the 2.5-year follow-up was 77% in patients without a persistent response and 90% in patients with a persistent response. At the 5-year follow-up, the DFS was 60% in patients without a persistent response, while it was 90% in patients with a persistent response (Figure 5C).

The OS at 2.5 years was almost 100% in endemic KS, 95% in patients with classic and epidemic types, and 65% in iatrogenic KS (Figure 6A). The DFS at the 2.5-year follow-up was 100% in patients with the iatrogenic type, 90% in patients with the epidemic type, 75% in the classic type, and 60% in the endemic type. At the 5-year follow-up, the DFS was nearly 70% in the patients with classic and endemic types, while it was 85% in the epidemic type and 100% in the patients with the iatrogenic type (Figure 6B).

The analysis of survival with a Cox regression model clearly shows that mortality is significantly higher in the iatrogenic type (0.011) and consequently, in transplanted subjects (0.008) (Table 2). The other significant finding (0.001) is that subjects with a persistent response have a significantly better disease-free survival.

## 4. Discussion

This study presents an analysis of the demographic and clinicopathological characteristics of a cohort consisting of 86 cases of KS, with the aim of expanding the knowledge about this rare form of cancer. Our findings show data of a KS cohort enrolled over almost two decades in a single center in northern Italy, a Mediterranean area with middle/low HHV-8 seroprevalence and inclusive of all four subtypes. An interesting and novel feature of our study resides in the analysis of the different factors related to the complex management of this sarcoma, which highlights the need for a multidisciplinary approach.

The demographics of our study confirmed the findings from previous studies [7]. In our case series, the average age at diagnosis was 56.9 years, confirming that KS is a disease that predominantly affects adults in their sixth decade. While KS typically exhibits a higher prevalence in men, our study observed a male-to-female ratio of 8.5:1, exceeding the ratio documented in the existing literature and probably explained by the high number of cases of classical and epidemic KS in our series [7]. Interestingly, the gender difference was attenuated in iatrogenic KS (male-to-female ratio of 3.25:1), thus confirming the data from a recent study about sex differences in cancer incidence among OTRs [12]. This study hypothesizes that immunosuppression in the setting of transplantation may balance cancer risk and decrease the male-to-female ratio reported in other subtypes [12]. 

The main risk factors for KS are HHV-8 infection and immunosuppression status [9]. Recently, chronic inflammation has also been recognized as a factor that plays an important role in the pathogenesis and prognosis of sarcomas [13].

Furthermore, C-reactive protein levels seem to have a significant positive correlation with poor prognosis [14]. 

A limitation of our study concerns the lack of data regarding HHV-8 status. The HHV-8 status was established in 55.81% of cases, but when tested, was positive in 48/51 (94.1%) of patients; we reported few cases with a negative immunohistochemistry for HHV-8 (3/51) but histologic confirmation of KS. Previous case reports suggested that in cases with a negative immunohistochemistry for HHV-8 but a clinical and histology compatibility with KS, it is useful to perform PCR to confirm the presence of HHV-8 [15]. The literature also suggests that negativity for HHV-8 may be associated with the dedifferentiation of the disease [16].

Patients with classical KS constituted the largest group in our study, supporting the evidence that Italy is one of the countries with the highest HHV-8 seroprevalence and incidence of sporadic/Mediterranean disease in Europe [17].

Patients with epidemic KS represented about one third of our sample. In HIV-positive patients, the mean age at diagnosis was generally lower, and disseminated disease with the involvement of skin, mucous membranes, lymph nodes, and viscera were more frequently observed.

An HHV-8 infection and low CD4 count are the main risk factors for developing the disease in HIV-positive patients, but given the retrospective nature of our study, it was difficult to retrieve complete data.

The introduction of combination antiretroviral therapy (cART) led to a remarkable decline in epidemic KS incidence and a significant improvement in KS prognosis due to the rise in CD4 T-cell count and immune reconstitution [18,19,20]. However, the risk of developing KS in HIV-positive patients with a normal CD4 T-cell count remains substantially higher than in the general population [21].

In our case series, the most widely used therapeutic strategy in HIV-positive patients was cART alone or combined with surgery, radiotherapy, and/or chemotherapy.

The literature data report that epidemic KS can regress following starting cART alone, although the response rates were variable [22,23].

In our case series, the iatrogenic KS group was mainly constituted by organ transplant recipients (OTRs), who had several comorbidities and a higher mortality (*p*-value 0.008), mainly unrelated to KS but to the inherent complexity and fragility of these patients. The high incidence of KS and high mortality in OTRs have been reported also by the literature [17,22,23]. Italian OTRS have an increased risk (about 100 times greater) for KS compared to the general population, especially during the first two years after transplantation [24]. Both an age over 30 at transplantation and a more aggressive immunosuppressive regimen were identified as independent risk factors for disease [25].

Interestingly, in our cohort, a decrease in cases of iatrogenic KS has been observed in the last decade. The decrease in the incidence of iatrogenic KS may be related to the introduction in clinical practice of new generation immunomodulatory drugs, which have led to an overall reduction in the immunosuppressive status of these patients [26]. In particular, the increasing use of mTOR inhibitors, which exhibit antitumoral and antiangiogenetic activities, may have played a role in the decline of KS occurrence in transplant recipients [27].

The comparison of the two most representative populations included in our study (HIV-positive patients vs. OTRs) suggests that the immunosuppression status has a significant impact on the KS patient’s response to therapy and prognosis. The population represented by OTRs is burdened with a high mortality likely related to comorbidities, but it is also strongly influenced by immunosuppression due to the chronic required immunosuppressive regimen. In HIV-positive patients, c-ART, by reducing immunosuppression, results in a benefit to these patients. Unfortunately, we have no comprehensive data on CD4 counts to support the link between the immunosuppressive status and prognosis of KS.

About 40% of our patients received multiple treatments. The high recurrence rate, which is an inherent feature of the disease, results in the need for a long treatment course consisting of several different approaches. In addition, the lack of treatment guidelines frequently poses difficulties in initiating a timely and effective therapeutic intervention.

Iatrogenic KS usually responds to immune reconstitution, though lowering the immunosuppressive dose risks graft rejection. Switching the chemical immune-suppressive regimen from cyclosporine A/FK506 to mammalian target of rapamycin (mTOR) inhibitors such as rapamycin/sirolimus/everolimus often leads to KS regression [27,28].

Epidemic KS responds to immune reconstitution and HIV suppression. The responses to cART alone are variable, and the complete remission of KS occurs in 20 to 80% of patients and is more common in cART-naïve patients with limited-stage disease compliant with HIV therapy [29,30]. The mechanism of KS regression is thought to be immunological and specifically associated with the reconstitution of an effective cytotoxic cell–mediated response targeting HHV-8. However, as KS can rapidly progress and affect vital organs, patients with advanced-stage KS need additional systemic anticancer treatments alongside cART [31,32]. There are no controlled trials to evaluate the best care of these patients, although systemic chemotherapy for KS and avoiding steroid treatment which was associated with disease progression seems appropriate, as reported in small series and case reports [33,34,35,36,37]. KS generally has a good prognosis. In our study, the 5-year OS was 83%, while the DFS was 80%, and no patient in our sample died from KS. The 5-year survival in our case series is slightly higher than that which is described in a recent epidemiological study on the U.S. population, which reports a 5-year relative survival in older adults (>40 years) of 77.5% and in adolescents and young adults (AYAs, 15–39 years) of 68.7%. The worse prognosis in AYAs may be explained by the higher incidence of HIV-associated KS among young patients [38]. 

Overall, comorbidities were very frequent in all patients with KS. Therefore, a multidisciplinary approach and the presence of multiple referral specialists is essential for the appropriate management of this disease.

Our study has some limitations, primarily related to its retrospective nature that makes it difficult to collect some important data including the HHV-8 status, HIV viral load, and CD4 lymphocyte counts. Because of the retrospective design of our study, missing data, and the small number of patients enrolled, our results should be interpreted with caution, and larger multicenter studies are needed.

The most interesting finding that has emerged concerns the complexity of these patients, suggesting the need to create an integrated multidisciplinary team capable of providing dedicated care to KS patients. The dermatologist could then assume the role of mediator among the various specialists in order to provide better patient care.

## 5. Conclusions

This study reflects our experience in the management of KS and provides valuable insights into KS characteristics, treatments, and outcomes. Comorbidities are very frequent in patients affected by KS, and despite the wide range of systematic and local treatment options, there is still no unified strategy for the treatment of KS. Therefore, a multidisciplinary approach for treating the patient with KS and the presence of multiple referral specialists are essential for the appropriate management of this disease during the diagnosis, treatment, and follow-up.

## Figures and Tables

**Figure 1 cancers-16-00691-f001:**
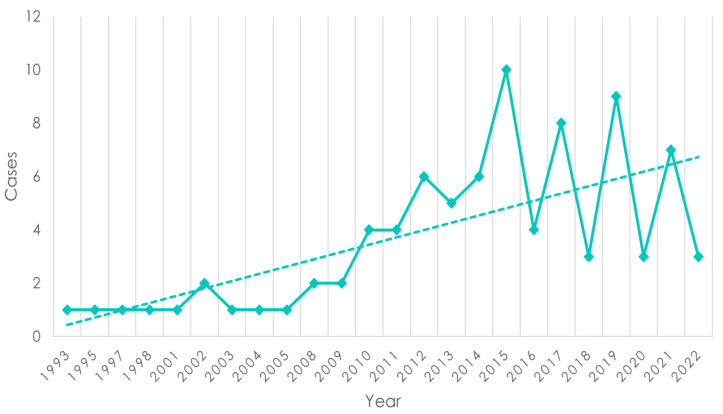
Temporal evolution of Kaposi’s sarcoma incidence: analysis of the time trend of new cases from 1993 to 2022. The dashed regression line was estimated using the ordinary least squares (OLS) method.

**Figure 2 cancers-16-00691-f002:**
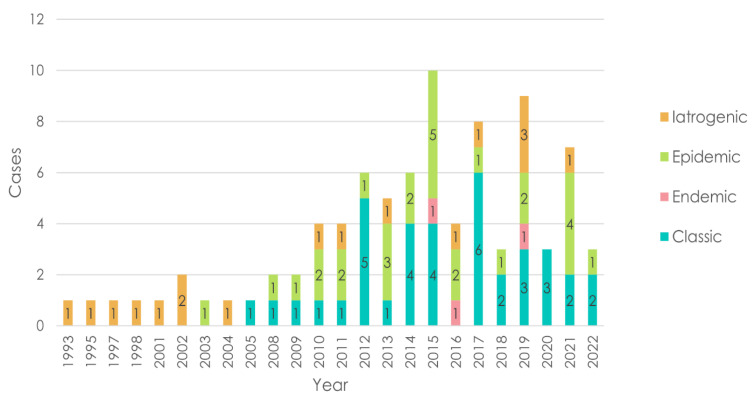
Temporal evolution of the distribution of Kaposi’s sarcoma types over the years.

**Figure 3 cancers-16-00691-f003:**
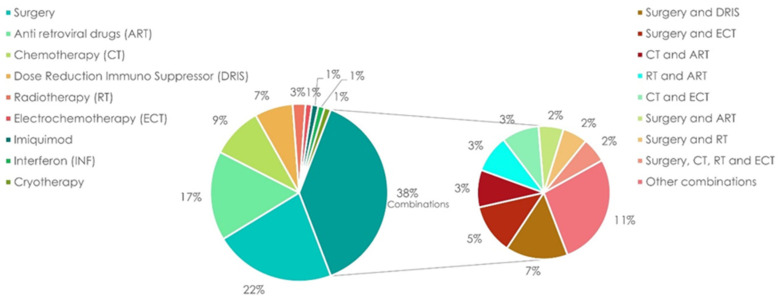
Kaposi’s sarcoma treatment distributions in our cohort.

**Figure 4 cancers-16-00691-f004:**
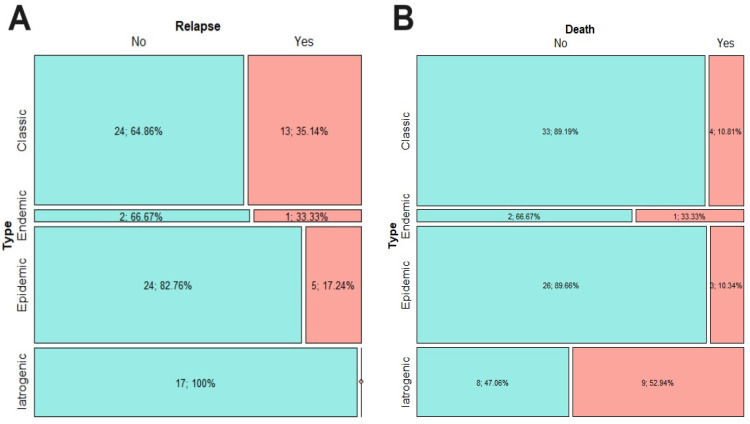
Relapse events in the 4 clinical–epidemiological types (**A**) and mortality in the four clinical–epidemiological types (**B**).

**Figure 5 cancers-16-00691-f005:**
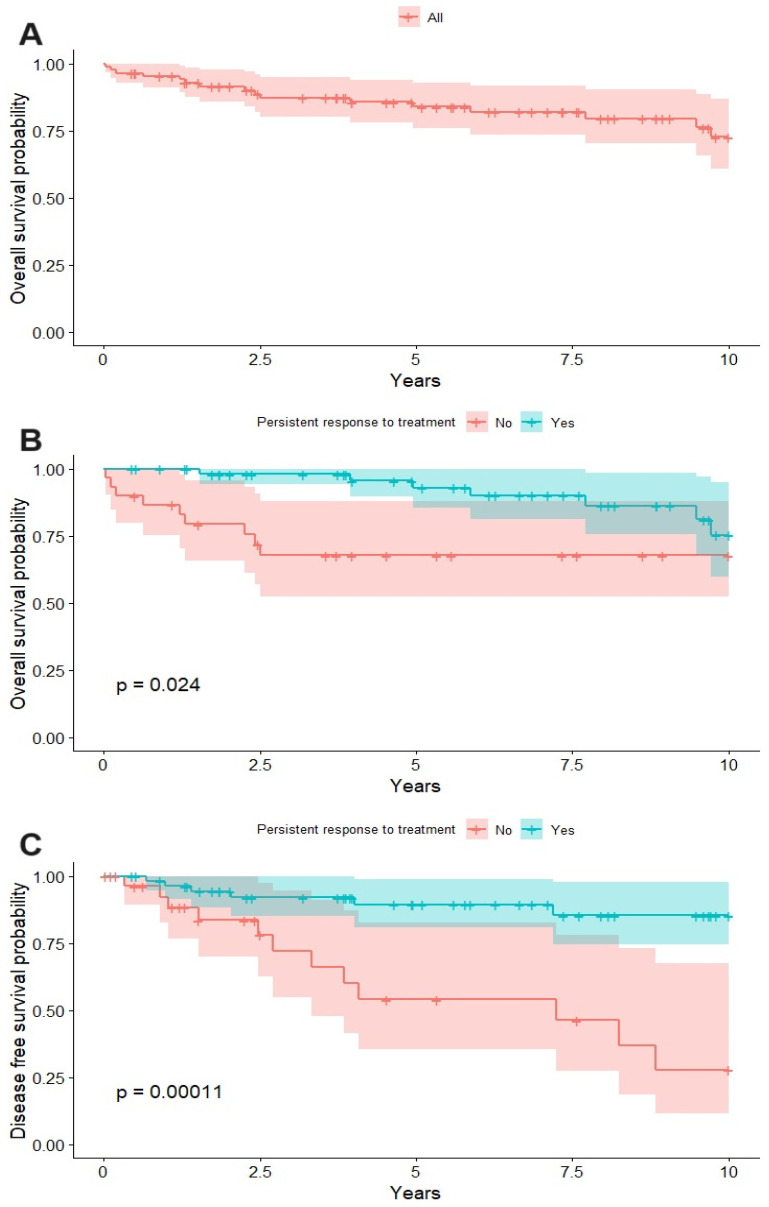
The 10-year survival curve (**A**), comparison of survival in persistent and nonpersistent response groups (**B**), and disease-free survival in the persistent response and nonpersistent response groups (**C**).

**Figure 6 cancers-16-00691-f006:**
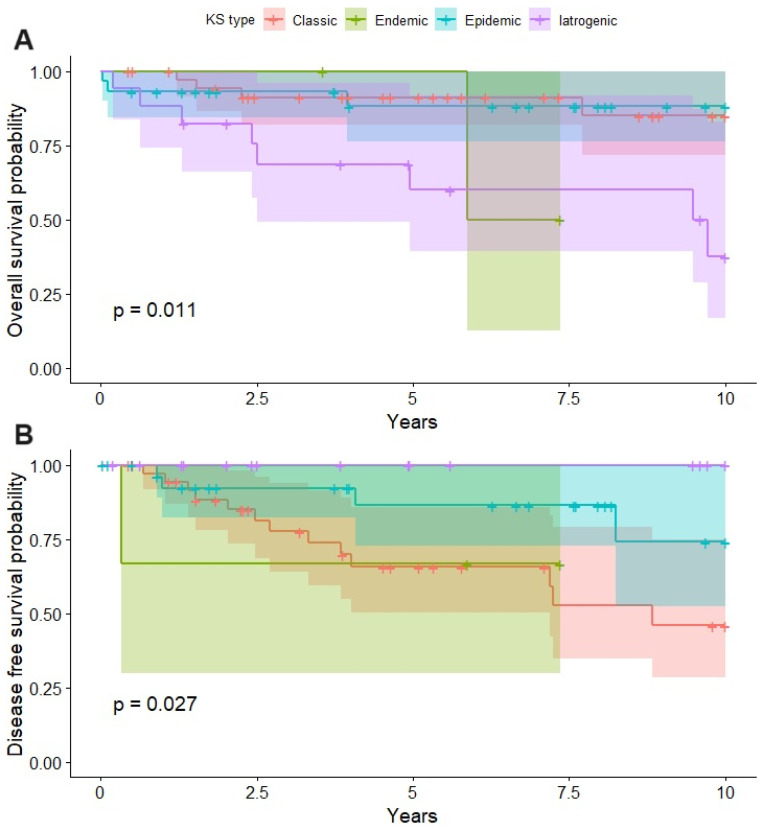
Overall survival for different types of KS (**A**) and disease-free survival for different types of KS (**B**).

**Table 1 cancers-16-00691-t001:** Demographic, tumor, and treatment characteristics of patients with a histologically confirmed Kaposi’s sarcoma who were treated at our institutions between 1993 and 2022.

Variable		Value
Age	Min/max	13.9/85.6
	Med [IQR]	57.2 [45.1; 69.2]
	Mean (std)	56.9 (14.7)
Gender	Woman	9 (10.47%)
	Man	77 (89.53%)
Tumor Site	Single site	53 (61.63%)
	Lower limbs	44 (51.16%)
	Head and neck	4 (4.65%)
	Upper limbs	3 (3.49%)
	Trunk	2 (2.33%)
	Multiple sites	33 (38.37%)
	Lower and upper limbs	8 (9.30%)
	Lower limbs and trunk	7 (8.14%)
	Lower and upper limbs, trunk, head, and neck	4 (4.65%)
	Lower and upper limbs, head, and neck	4 (4.65%)
	Lower limbs, trunk, head, and neck	3 (3.49%)
	Lower and upper limbs and trunk	3 (3.49%)
	Other combinations	4 (4.65%)
Type of KS	Classic	37 (43.02%)
	Endemic	3 (3.49%)
	Epidemic	29 (33.72%)
	Iatrogenic	17 (19.77%)
Involvement	Only cutaneous	67 (77.91%)
	Cutaneous and extracutaneous (node or mucosal)	14 (12.28%)
	Only extracutaneous (node or mucosal)	5 (5.81%)
HIV	No	56 (65.12%)
	Yes	30 (34.88%)
Transplant	Yes	13 (15.12%)
	No	73 (84.88%)
Comorbidity	No	12 (13.95%)
	Yes	74 (86.05%)
HHV-8 Status	ND	35 (39.53%)
	No	3 (3.37%)
	Yes	48 (55.81%)
Treatment	Single	53 (61.63%)
	Multiple	33 (38.37%)
Persistent Response	No	30 (34.88%)
	Yes	56 (65.12%)
Relapse	No	67 (77.91%)
	Yes	19 (22.09%)
Deceased	No	69 (80.23%)
	Yes	17 (19.77%)
Cause of Death	COPD or respiratory failure	5 (%)
	Cardiac	1 (5.88%)
	Infection, pneumonia, or sepsis	3 (%)
	Liver failure	4 (23.53%)
	Chronic renal failure	1 (5.88%)
	Neoplasm	2 (11.76%)
	Of old age	1 (5.88%)
Follow-up Time (Years)	Min/max	0.02/30.1
	Med [IQR]	5.8 [2.4; 9.7]
	Mean (std)	6.6 (5.1)
Time to Relapse (Years)	Min/max	0.3/15.0
	Med [IQR]	2.7 [1.2; 5.6]
	Mean (std)	4.0 (3.8)
Time to Death (Years)	Min/max	0.02/11.1
	Med [IQR]	2.4 [1.2; 5.9]
	Mean (std)	3.8 (3.7)

**Table 2 cancers-16-00691-t002:** Cox regression analysis.

Variable		Overall Survival		DFS	
		1/HR	*p*-Value	1/HR	*p*-Value
Age		0.993	0.689	0.973	0.137
Gender	Man (ref woman)	0	0.998	0	0.998
Tumor Site	Lower limbs (present, ref absent)	1.136	0.916	0.428	0.421
	Head and neck (present, ref absent)	2.103	0.533	2.780	0.213
	Upper limbs (present, ref absent)	1.773	0.461	0.731	0.568
	Trunk (present, ref absent)	4.563	0.152	0.482	0.179
Type of KS	Endemic (ref classic)	0.298	0.282	0.958	0.967
	Epidemic (ref classic)	1.015	0.985	2.758	**0.077**
	Iatrogenic (ref classic)	0.211	**0.011**	>>10	0.998
Involvement	Cutaneous and extracutaneous (ref only cutaneous)	1.466	0.613	0.298	**0.017**
	Only extracutaneous (ref only cutaneous)	>>10	0.998	>>10	0.998
HIV	Yes (ref no)	2.308	0.192	1.977	0.23
Transplant	Yes (ref no)	0.254	**0.008**	>>10	0.997
Comorbidity	Yes (ref no)	0	0.997	0.351	0.309
Autoimmune/Immunologic	ND (ref no)	3.401	0.244	0	0.998
	Yes (ref no)	3.220	0.314	0	0.998
Neoplastic	ND (ref no)	2.653	0.358	2.636	0.362
	Yes (ref no)	1.478	0.719	5.022	0.172
Treatment	Surgery (undergone, ref not)	3.630	**0.052**	0.759	0.649
	Antiretroviral drugs (undergone, ref not)	2.747	0.171	1.942	0.429
	Chemotherapy (undergone, ref not)	1.092	0.897	0.480	0.226
	Dose reduction of immunosuppressive therapy (undergone, ref not)	0.930	0.918	>>10	0.998
	Radiotherapy (undergone, ref not)	3.296	0.252	0.526	0.263
	Electrochemotherapy (undergone, ref not)	2.593	0.372	0.577	0.336
Persistent Response	Yes (ref no)	2.958	**0.032**	5.616	**0.001**

HR: Hazard ratio (1/HR >1 indicates better survival than the reference class, <1 indicates worse); Bold formatting is used for *p*-value < 0.10 to better visualize statistically significant variables.

## Data Availability

The datasets presented in this study can be found in online repositories. The name of the repository/repositories and accession number(s) can be found here: https://zenodo.org/records/10473499 (accessed on 26 January 2024).

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
