# Peer review of "Kaposi’s Sarcoma: Evaluation of Clinical Features, Treatment Outcomes, and Prognosis in a Single-Center Retrospective Case Series"

_cancers, 2024, doi:10.3390/cancers16040691_

Round 1
Reviewer 1 Report
Comments and Suggestions for Authors
The authors show the prescious data of which Kaposi’s Sarcoma.
The study is interesting, however, I have some concerns to discuss.
-What is the novelty of the current study?
-How does inflammation involved in the pathogenesis of kaposi sarcoma?
Please discuss referring the following article.
-Characterizing inflammatory markers in highly aggressive soft tissue sarcomas. Medicine (Baltimore). 2022 Sep 30;101(39):e30688. doi: 10.1097/MD.0000000000030688. PMID: 36181081; PMCID: PMC9524936.
Author Response
The authors show the precious data of which Kaposi’s Sarcoma.
The study is interesting, however, I have some concerns to discuss.
1)What is the novelty of the current study?
RE: Thank you for your comment. The novelty of our study is that it was performed on a single centre cohort in Northern Italy, an area characterized by a middle/low seroprevalence of HHV-8, and it includes all four different clinico-epidemiological subtypes of KS. We highlight both clinical and prognostic differences among the various forms and the complexity of the management, which should imply a multidisciplinary approach. This is now included in the manuscript (page 10, lines 217-221).
Moreover, limited data exist regarding the temporal evolution in the incidence of the four epidemiological forms of KS. In our study, we analyzed data over a long period of time to identify differences in temporal patterns. In our cohort, a notable decrease in cases of iatrogenic KS has been observed over the last decade. This decline in the incidence of iatrogenic KS may be attributed to the introduction of new generation immunomodulatory drugs in clinical practice. These advancements have contributed to an overall reduction in the immunosuppressive status of these patients. This potentially novel finding has already been discussed in the manuscript (page 11, lines 271-277).
2) How is inflammation involved in the pathogenesis of kaposi sarcoma? Please discuss referring the following article.
Characterizing inflammatory markers in highly aggressive soft tissue sarcomas. Medicine (Baltimore). 2022 Sep 30;101(39):e30688. doi: 10.1097/MD.0000000000030688. PMID: 36181081; PMCID: PMC9524936.
RE: Thank you for your suggestion; we have now briefly discussed the role of inflammation in KS pathogenesis and included this interesting paper (pages 10-11, lines 233-236, and new references 12 and 13).

Reviewer 2 Report
Comments and Suggestions for Authors
The manuscript represents a general analysis of the incidence of Kaposi sarcoma, treatment and outcome in a location in Italy. It is a general, not original research, although it refers to a particular population. The information is quite general, and the most relevant information is in Figures 5 and 6, particularly Figure 5. The authors did not include the limitations of the study, which may be referred to as the analysis within the guidelines in treatment that may be applied. The discussion should compare some other studies with different ethnicity and age see DOI: 10.1200/JCO.23.01367 and also solid organ recipients DOI: 10.1093/jnci/djad224
Comments on the Quality of English LanguageMinor grammatical mistakes were encountered
Author Response
SPECIFIC ANSWERS TO REVIEWERS’ COMMENTS
Reviewer #2
1) The manuscript represents a general analysis of the incidence of Kaposi sarcoma, treatment and outcome in a location in Italy. It is a general, not original research, although it refers to a particular population. The information is quite general, and the most relevant information is in Figures 5 and 6, particularly Figure 5. The authors did not include the limitations of the study, which may be referred to as the analysis within the guidelines in treatment that may be applied. The discussion should compare some other studies with different ethnicity and age see DOI: 10.1200/JCO.23.01367 and also solid organ recipients DOI: 10.1093/jnci/djad224
RE: Thank you for your constructive comment. We revised our manuscript according to your comments; we have now compared our findings with the studies suggested (page 10, lines 227-231; page 12, lines 306-310, references 11 and 37).
Minor grammatical mistakes were encountered
We have now rechecked and edited the manuscript accordingly.

Round 2
Reviewer 1 Report
Comments and Suggestions for Authors
The authors replied well, so the manuscript is suitable for publication.
Author Response
Thank you for your comments, your revision suggestions have significantly improved our manuscript.
Best regards
Paolo
Reviewer 2 Report
Comments and Suggestions for Authors
The manuscript was improved. It is now suitable for publication
Comments on the Quality of English LanguageMinor grammatical mistakes were observed.
Author Response
Thank you for your comments, your revision suggestions have significantly improved our manuscript.